# SparseEval: Efficient Evaluation of Large Language Models by Sparse Optimization

**Taolin Zhang**[1,2]**, Hang Guo**[2]**, Wang Lu**[3†]**, Tao Dai**[1†]**, Shu-Tao Xia**[2]**, Jindong Wang**[4]

[1]College of Computer Science and Software Engineering, Shenzhen University
[2]Tsinghua Shenzhen International Graduate School, Tsinghua University
[3]Independent, [4]Microsoft Research Asia
{newlw230630, daitao.edu}@gmail.com

## Abstract

As large language models (LLMs) continue to scale up, their performance on various downstream tasks has significantly improved. However, evaluating their capabilities has become increasingly expensive, as performing inference on a large number of benchmark samples incurs high computational costs. In this paper, we revisit the model-item performance matrix and show that it exhibits sparsity, that representative items can be selected as anchors, and that the task of efficient benchmarking can be formulated as a sparse optimization problem. Based on these insights, we propose SparseEval, a method that, for the first time, adopts gradient descent to optimize anchor weights and employs an iterative refinement strategy for anchor selection. We utilize the representation capacity of MLP to handle sparse optimization and propose the Anchor Importance Score and Candidate Importance Score to evaluate the value of each item for task-aware refinement. Extensive experiments demonstrate the low estimation error and high Kendall's $\tau$ of our method across a variety of benchmarks, showcasing its superior robustness and practicality in real-world scenarios. Code is available at https://github.com/taolinzhang/SparseEval.

## 1 Introduction

In recent years, the capabilities of large language models (LLMs) have improved dramatically as model scales have grown (Brown, 2020; Achiam et al., 2023; Bai et al., 2023; Yang et al., 2024; Team, 2025; Guo et al., 2025). From early small- and medium-sized models to today's hundred-billion-parameter giants, LLMs have achieved remarkable performance in natural language understanding, reasoning, and generation tasks. However, this performance gain comes along with the increased inference cost and evaluation overheads. Larger models require significantly more computational resources and time for inference, particularly during evaluation where the cost of running on large-scale datasets may become prohibitively high. This raises a crucial question in both research and practical deployment: how can we reduce inference costs while maintaining evaluation quality?

Previous studies have noted that evaluation datasets often contain a significant amount of redundancy, as demonstrated by visualization techniques such as UMAP (McInnes et al., 2018) and t-SNE (Maaten and Hinton, 2008). These approaches analyze similarities between samples based on input prompts or probability distributions, revealing underlying patterns in the data. However, such methods typically rely on prompts or predicted probabilities to detect redundancy, which requires additional resources that may be costly or difficult to acquire. Building on these observations, recent work employs Item Response Theory to reduce the number of evaluation samples (Polo et al., 2024) , while others utilize adaptive clustering strategies for data selection (Yuan et al., 2025; Wang et al., 2025).

Building such an efficient evaluation benchmark necessitates addressing three core challenges. First is **sparsity existence**: how can we intuitively and quantitatively demonstrate the presence of sparsity in evaluation tasks to justify efficient evaluation? Second is **anchor weighting optimization**: given pre-selected anchors, how can we optimize their weights so that they effectively represent the

---

†Corresponding author.

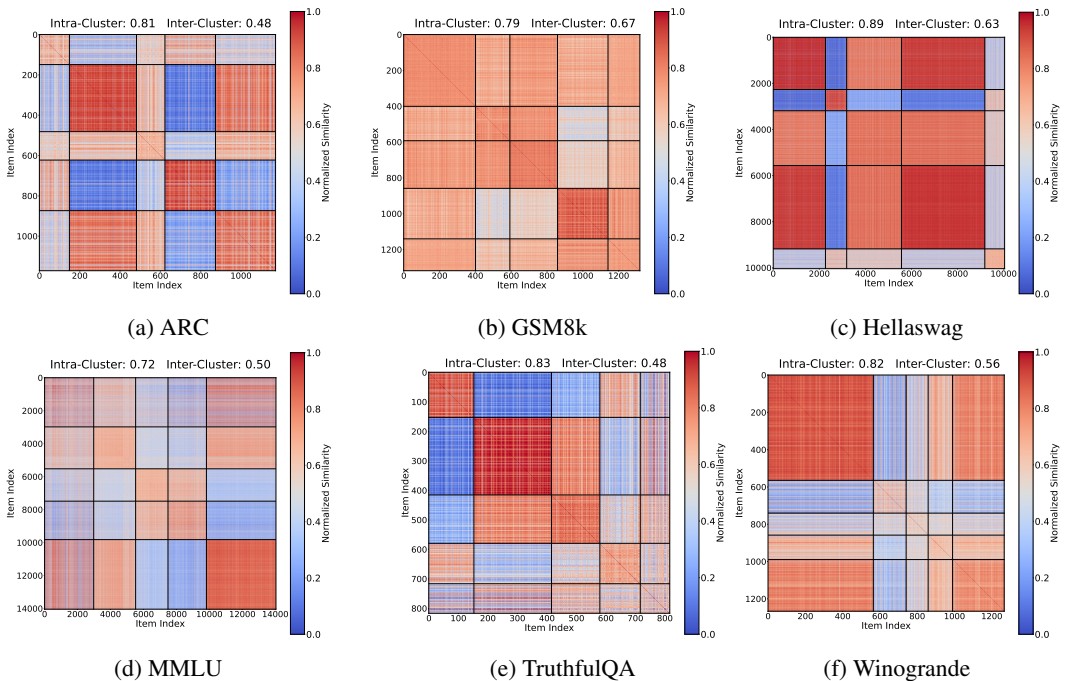

Figure 1: **Evidence of Evaluation Sparsity in LLM Benchmarks.** We construct an item-item similarity matrix by computing the cosine similarity between item vectors. The presence of pronounced diagonal blocks along with both high intra- and inter-cluster similarity, suggests the existence of evaluation sparsity and redundancy in the benchmark.

characteristics of the entire dataset? Third is **anchor selection**: how should the choice of anchors be guided by the optimized weights and downstream tasks to ensure higher task relevance in evaluation results? These challenges form the backbone of designing an effective evaluation framework.

In this work, we revisit the *model-item matrix* and use *spectral clustering* to uncover its highly structured and sparse nature. Our analysis reveals that even clustering data with a few anchor points leads to a high degree of intra-cluster similarity and strong inter-cluster predictability. This suggests that the model predictions across samples encode a large amount of transferable information, providing support for efficient evaluation. Therefore, we propose **SparseEval**, a task-aware efficient evaluation for large foundation models by sparse optimization. For anchor weight optimization, we propose a training strategy based on gradient descent optimization and dynamically adjusts weights by optimizing reconstruction loss. For anchor selection, we design a task-aware strategy that leverages error-correlation anchor refinement to identify more representative anchor subsets. SparseEval not only reduces evaluation cost but also enhances alignment between the anchors and downstream task.

Unlike existing methods, our framework capture evaluating sparsity directly from the model-item matrix without relying on additional embeddings. By integrating spectral clustering with gradient-based optimization, our approach accurately predicts model performance using a minimal number of anchor points. This results in a practical and scalable solution that strikes a strong balance between computational efficiency and evaluation accuracy. Our experiments show that compared to traditional full-dataset evaluation, our method can reduce inference costs to only 100 instances while maintaining low estimate errors and highly consistent performance estimates. Moreover, our framework is highly generalizable and can be easily adapted to various evaluation settings and task types, offering a new direction for designing efficient evaluation benchmarks in the era of large-scale language models.

## 2   RELATED WORKS

**Efficient Data Selection**   Various studies have explored efficient data selection across different settings. LIMA (Zhou et al., 2023) demonstrates that a large language model can be effectively

aligned using only 1,000 high-quality examples, as most of the model's knowledge is obtained during pretraining. LIMR (Li et al., 2025) argues that in the context of reinforcement learning for LLM training, a small but strategically curated dataset can outperform significantly larger ones. LIMO (Ye et al., 2025) further shows that strong mathematical reasoning capabilities can be elicited from a knowledge-rich LLM by fine-tuning it on just 800 high-quality examples, surpassing models trained on datasets more than 100 times larger. These approaches primarily aim to extract high-quality samples from the original dataset, rather than aligning performance on the original dataset.

**Efficient LLM Evaluation** Evaluating LLMs is often resource intensive due to the large number of models and test items involved. Several methods have been proposed to reduce evaluation costs by selecting a subset of models or items, while closely aligning with the original benchmark in terms of both accuracy and ranking—a field often referred to as Efficient LLM Evaluation. Anchor Points (Vivek et al., 2023) analyzes language models using small and representative subsets of evaluation data, while Flash-HELM (Perlitz et al., 2023) adopts a coarse-to-fine strategy that adaptively adjusts the number of evaluation samples for higher-ranking models. In addition, Pacchiardi et al. (2024) proposes using performance on a small set of reference instances as input features to a general assessor model to predict a new LLM's performance on unseen instances. TinyBenchmark (Polo et al., 2024) employs Item Response Theory (IRT) to guide data selection and TailoredBench (Yuan et al., 2025) presents another approach by creating a customized, adaptive benchmark tailored to each model. Recently, EffiEval (Wang et al., 2025) aims to improve evaluation efficiency by selecting a small subset of data that maximizes the coverage of a model's internal capabilities.

# 3 LLM EVALUATION CAN BE SPARSE

In this section, we first discuss the evalution sparsity in current benchmarks by revisiting the model-item performance matrix. Next, we formally define the problem formulation of efficient evaluation as sparse optimization.

## 3.1 EVALUATING SPARSITY IN LLM BENCHMARKS

Given a model-item score matrix $S \in \{-1, 1\}^{m \times n}$, where $m$ and $n$ denote the number of models and items respectively, and where $S_{i,j} = 1$ indicates a correct prediction and $S_{i,j} = -1$ indicates an incorrect one. We observe an inherent sparsity structure in this evaluation matrix, which enables more efficient evaluation. In particular, we analyze inter-item relationships by examining the column vectors of $S$, where each column represents the response patterns of all models to a specific item. By computing the cosine similarity between these item vectors, we construct an item-item similarity matrix $S_{\text{item}} \in \mathbb{R}^{n \times n}$, where each entry reflects the similarity between a pair of test items.

To further explore the structure of the dataset, we apply spectral clustering to $S_{\text{item}}$ and partition the test items into five clusters, as shown in Figure 1. The resulting clusters reveal strong intra-cluster similarity, as evidenced by the pronounced diagonal blocks in the clustered similarity matrix. This observation indicates that many test items are highly similar to each other in terms of model response patterns, suggesting the presence of redundancy and potential for sparsity in the benchmark. Such a clustering structure implies that it may be feasible to select a small subset of representative items (referred to as **anchors**) that can effectively capture the diversity of the entire dataset. Moreover, we observe that there also exists considerable inter-cluster similarity among the items after clustering, indicating that all the item vectors share substantial common information, making mutual prediction possible. This reveals the widespread existence of **Evaluation Sparsity** in LLM benchmarks. By focusing evaluation on these anchors, we can potentially reduce the overall evaluation cost while preserving the discriminative power of the benchmark, which motivates the subsequent development of anchor-based sparse evaluation.

## 3.2 PROBLEM FORMULATION

The goal is to reduce the evaluation cost of LLM benchmarks by selecting a small subset of informative items, such that performance evaluated on this subset can closely approximate the result on the full dataset. We achieve this by learning a item-level weighting function, and cast this as an sparse optimization problem.

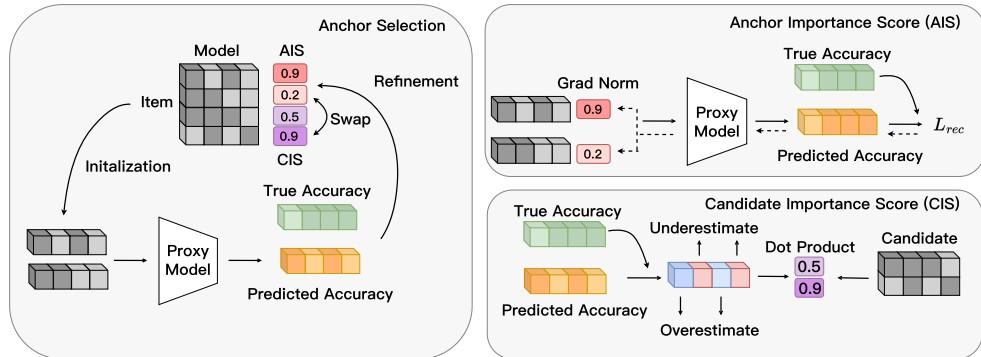

Figure 2: **Anchor Refinement in SparseEval.** We leverage a proxy model to perform task-aware anchor refinement. By iteratively replacing items with low Anchor Importance Scores with those having high Candidate Importance Scores, we are able to obtain more representative anchors for efficient evaluation.

**General Formulation** Let $S = [s_1, s_2, \ldots, s_n] \in \mathbb{R}^{m \times n}$ denote the per-item evaluation scores. With given sparsity $k$, we construct the sparse input $S' = S \odot (\mathbf{1}_m W^\top)$ by introducing weighting factor $W \in \mathbb{R}^n$ ($\|W\|_0 \le k$), where $\odot$ denotes the element-wise product. Let $f : \mathbb{R}^n \to \mathbb{R}$ be an aggregation function that maps the sparse input to an overall benchmark score. We now formulate the evaluation task as the following optimization problem:

$$
\begin{aligned}
\underset{W, f}{\text{minimize}} \quad & \left| f(S \odot (\mathbf{1}_m W^\top)) - SW_a \right|_1, \\
\text{subject to} \quad & \|W\|_0 \le k
\end{aligned}
\tag{1}
$$

where $W_a = \frac{1}{n}\mathbf{1}_n$ is a uniform averaging vector over the full set of $n$ items and $SW_a$ denotes the real performance of the model. This formulation encourages the learned function $f(S \odot (\mathbf{1}_m W^\top))$ applied to a small subset of items to approximate the full evaluation $W_a^T S$.

**Linear Formulation** When $f$ is a linear function, it can be absorbed by $W$ and we can derive the objective of all previous methods (Polo et al., 2024; Yuan et al., 2025), which directly selects a sparse subset of items whose weighted average matches the full benchmark score.

$$
\begin{aligned}
\underset{W}{\text{minimize}} \quad & \|SW - SW_a\|_1 \\
\text{subject to} \quad & \|W\|_0 \le k
\end{aligned}
\tag{2}
$$

## 4 METHODOLOGY

In this section, we first describe the anchor weight predictor for pre-selected anchors in an end-to-end optimization. We then discuss the initialization and refinement strategy of anchors in SparseEval.

### 4.1 ANCHOR WEIGHT PREDICTOR

We begin by focusing on a simplified version of the problem: given a fixed set of anchor points, we aim to find an appropriate method for assigning weights to these anchors for perfomance estimation. Traditional methods, such as TinyBenchmark (Polo et al., 2024), typically train an Item Response Theory (IRT) model and then apply $k$-means clustering to the resulting IRT representations in order to identify anchor points. The weights of these anchors are subsequently assigned based on the distribution of the clusters. Other approaches (Yuan et al., 2025) leverage scaling factors to calibrate the performance estimation. In contrast, by formulating the task as a sparse optimization problem, we observe that the aggregation function $f$ can be approximated using a MLP model for end-to-end optimization. Therefore, it becomes possible to directly optimize the anchor weights via gradient descent by minimizing a reconstruction loss.

---

**Algorithm 1** SparseEval

---

**Require:** Prediction matrix $S \in \{-1, 1\}^{m \times n}$, number of anchors $k$, refinement steps $R$, learning rate $\eta$, proxy training epochs $E$, final training epochs $F$
 1: **Anchor Selection and Refinement:**
 2: Initialize anchors $A$ using $k$-means or random sampling based on validation set
 3: **for** $r = 1 \rightarrow R$ **do**
 4:     Train a proxy MLP $f_r$ on current anchors $A$ with $E$ epochs to predict overall model performance
 5:     Compute prediction residuals: $e$ in Eq. 4
 6:     **for** each anchor $i \in A$ **do**
 7:         Compute Anchor Importance Score $AIS_i$ in Eq. 6 for anchor $i$
 8:     **end for**
 9:     **for** each candidate $j \notin A$ **do**
10:         Compute Candidate Importance Score $CIS_j$ in Eq. 5 for candidate $j$
11:     **end for**
12:     Remove weakest anchor $i^* = \arg\min_i \text{AIS}_i$ from $A$
13:     Add strongest candidate $j^* = \arg\max_j \text{CIS}_j$ to $A$
14: **end for**
15: **Final MLP Training:**
16: Train final MLP model $f$ on selected anchors $A$ with $F$ epochs to predict overall model performance
17: Return final trained MLP model $f$ and anchor set $A$

---

Formally, let $W$ denote the given sparse weight factor ($\|W\|_0 \leq k$), $W_a$ the ground-truth uniform averaging weights, and $S_{train} \in \mathbb{R}^{M \times n}$ the training score matrix over $M$ examples and $n$ items. The square loss function for anchor weight optimization is defined as:

$$\mathcal{L} = \frac{1}{M} \left\| f(S_{train} \odot (\mathbf{1}_M W^\top)) - S_{train} W_a \right\|_2 \tag{3}$$

## 4.2 ANCHOR INITIALIZATION

Based on our earlier observations, we find that most datasets exhibit strong intra-cluster similarity, which motivates us to initialize anchor points directly by $k$-means clustering. Applying $k$-means directly captures the cluster structure of the original dataset, leading to the selection of representative anchors that work well with subsequent gradient-based optimization. However, for datasets with weaker intra-cluster similarity such as MMLU, we observe that random initialization can also perform surprisingly well, and in some cases, it even outperforms $k$-means initialization.

Therefore, in practice, we adopt an adaptive strategy for anchor initialization. For each dataset, we evaluate both $k$-means and random initialization on a small set of models as the validation set and choose the better method for the final initialization.

## 4.3 ANCHOR REFINEMENT

Directly optimizing the weights of the anchors initialized as described above presents a key limitation that these initialization strategies are not tailored to the downstream task. The random initialization selects arbitrary points from the dataset, while the k-means approach captures only the clustering structure of the data. However, neither method participates in the end-to-end optimization process based on reconstruction loss, and the anchors remain fixed after initialization. As a result, the selected subsets may be suboptimal for perfomance estimation tasks.

To address this, we explore the relationship between anchors and candidates within the MLP model for anchor refinement. In our setup, the MLP takes the predictions of the anchors as input and uses the predictions of the candidates as targets to learn from. Two essential components naturally emerge from end-to-end training with reconstruction loss and backpropagation: *errors* and *gradients*. The error can be attributed to each non-anchor individually, reflecting how well the current set of anchors

predicts each non-anchor. Conversely, the gradients can be obtained by propagating the errors back through the network and can be assigned to each anchor, indicating its contribution and importance in the perfomance estimation.

Formally, given $m$ models, $n$ items, and a prediction matrix $S \in \{-1, 1\}^{m \times n}$, the model-level calibration residuals are computed as:

$$e = f(S \odot (\mathbf{1}_m W^\top)) - SW_a. \tag{4}$$

Intuitively, if $e_j > 0$, it means that the proxy model overestimates model $j$. In contrast, if $e_j < 0$, it underestimates it. Now, consider the prediction pattern of a candidate item $i$ as a feature vector, representing the model performance over this item. If this feature tends to be positive where the residual vector is positive and negative where it is negative, then the absolute value of their dot product will be high. That is, features that are aligned with the structure of the residuals can consistently indicate whether a model is being overestimated or underestimated. For example, if a feature value of $+1$ typically corresponds to overestimated models and $-1$ to underestimated ones, the dot product with the residuals will have a large absolute magnitude, suggesting that it is an informative feature. Based on this intuition, we define the **Candidate Importance Score (CIS)** for candidate item $i$ as the absolute value of the dot product between its prediction pattern and the residual vector:

$$\text{CIS}_i = \left| (S_{:,i})^T e \right| = \left| (S_{:,i})^T \left( f(S \odot \mathbf{1}_m W^\top) - SW_a \right) \right| \tag{5}$$

Simultaneously, we can leverage the magnitude of the gradient during backpropagation to assess the influence of a given anchor on optimizing the prediction error. This is especially useful in the context of fast-training proxy models, where the absolute gradient is often a more direct indicator of impact on the error than the weight activation from the first layer. Therefore, we can compute the **Anchor Importance Score (AIS)** for anchor $i$ as follows:

$$\text{AIS}_i = \left\| \frac{\partial \mathcal{L}}{\partial S_{:,i}} \right\|_1 = \frac{1}{N} \cdot \frac{\partial}{\partial S_{:,i}} \left\| f(S \odot \mathbf{1}_m W^\top) - SW_a \right\|_2. \tag{6}$$

At each step of the iterative refinement process, we apply gradient descent with a proxy MLP model on the current set of anchors to calculate AIS of each anchor and CIS of each candidate. We then replace the anchor with the lowest AIS with the candidate that has the highest CIS. After $R$ refinement steps, we train a final MLP model which sharing the same architecture but with a different number of input features as the final model.

## 4.4 THEORETICAL ANALYSIS

**Proposition 1** (More anchors yield no larger reconstruction error). *In linear weight setting, let $S \in \mathbb{R}^{m \times n}$ be the model–sample score matrix and define the true overall average as*

$$\mu = SW_a = \frac{1}{n} S\mathbf{1}_n \in \mathbb{R}^m,$$

*where $W_a = \frac{1}{n}\mathbf{1}_n$ is the uniform weight vector. For any anchor set $A \subseteq \{1, \ldots, n\}$, define the feasible linear weight class*

$$\mathcal{W}(A, k) := \{w \in \mathbb{R}^n : \text{supp}(w) \subseteq A, \|w\|_0 \leq k\}$$

*and the optimal reconstruction error*

$$E(A) := \min_{w \in \mathcal{W}(A,k)} \|Sw - \mu\|_1.$$

*Whenever $A \subseteq B$, it holds that*

$$E(B) \leq E(A).$$

**Proposition 2** (Anchor refinement decreases $L_2$ reconstruction error). *In the linear weight setting, consider the linear reconstruction model $\hat{\mu} = Sw$ with L2 loss $\mathcal{L}(w) = \|Sw - \mu\|_2^2$. Let $A$ be the current anchor set and $w$ the current linear weights. Let the residual be $r = Sw - \mu$. In linear weight setting, we have*

$$\text{CIS}_j = |s_j^\top r|, \qquad j \notin A,$$

Table 1: **Main results on LLM benchmarks.** SparseEval consistently outperform baselines by up to 2% lower estimate errors and 0.1 improvement in Kendall's $\tau$ than baselines.

| Dataset | Method | Anchor = 20 | | Anchor = 40 | | Anchor = 60 | | Anchor = 80 | | Anchor = 100 | |
|---|---|---|---|---|---|---|---|---|---|---|---|
| | | MAE (%) ↓ | $\tau$ ↑ | MAE (%) ↓ | $\tau$ ↑ | MAE (%) ↓ | $\tau$ ↑ | MAE (%) ↓ | $\tau$ ↑ | MAE (%) ↓ | $\tau$ ↑ |
| ARC | Anchor Points | 4.004 | 0.769 | 2.375 | 0.866 | 2.890 | 0.867 | 2.289 | 0.868 | 10.620 | 0.578 |
| | gp-IRT | 5.332 | 0.641 | 3.642 | 0.698 | 2.959 | 0.758 | 2.612 | 0.761 | 2.274 | 0.787 |
| | TailoredBench | 3.426 | 0.824 | 2.646 | 0.854 | 2.448 | 0.852 | 2.816 | 0.862 | 2.413 | 0.873 |
| | SparseEval | **1.778** | **0.863** | **1.581** | **0.883** | **1.404** | **0.902** | **1.227** | **0.910** | **1.165** | **0.917** |
| GSM8K | Anchor Points | 4.433 | 0.844 | 3.756 | 0.878 | 3.631 | 0.916 | 2.778 | 0.906 | 5.295 | 0.842 |
| | gp-IRT | 5.275 | 0.802 | 3.984 | 0.832 | 3.161 | 0.871 | 2.774 | 0.880 | 2.424 | 0.887 |
| | TailoredBench | 5.412 | 0.833 | 4.157 | 0.885 | 4.271 | 0.892 | 4.003 | 0.900 | 4.203 | 0.912 |
| | SparseEval | **3.305** | **0.872** | **2.321** | **0.908** | **1.960** | **0.925** | **1.754** | **0.931** | **1.619** | **0.936** |
| HellaSwag | Anchor Points | 3.272 | 0.796 | 2.619 | 0.875 | 2.416 | 0.856 | 1.962 | 0.847 | 2.012 | 0.889 |
| | gp-IRT | 5.323 | 0.661 | 3.501 | 0.687 | 2.754 | 0.745 | 1.992 | 0.784 | 1.750 | 0.783 |
| | TailoredBench | 2.352 | **0.811** | 2.257 | 0.847 | 1.868 | 0.861 | 1.957 | 0.857 | 1.968 | 0.876 |
| | SparseEval | **1.477** | 0.857 | **1.210** | **0.890** | **0.993** | **0.906** | **0.942** | **0.910** | **0.827** | **0.918** |
| MMLU | Anchor Points | 4.898 | 0.727 | 2.830 | 0.801 | 2.331 | 0.850 | 1.964 | 0.856 | 7.890 | 0.764 |
| | gp-IRT | 5.940 | 0.569 | 3.802 | 0.692 | 3.190 | 0.710 | 2.537 | 0.798 | 2.202 | 0.829 |
| | TailoredBench | 4.046 | 0.755 | 2.677 | 0.845 | 2.421 | 0.857 | 2.216 | 0.876 | 2.019 | 0.862 |
| | SparseEval | **1.718** | **0.832** | **1.282** | **0.871** | **0.997** | **0.890** | **0.962** | **0.896** | **0.842** | **0.908** |
| TruthfulQA | Anchor Points | 3.215 | 0.803 | 2.443 | 0.838 | 1.958 | 0.870 | 1.758 | 0.885 | 1.733 | 0.891 |
| | gp-IRT | 4.452 | 0.712 | 3.032 | 0.771 | 2.250 | 0.823 | 1.973 | 0.836 | 1.808 | 0.847 |
| | TailoredBench | 2.554 | 0.847 | 2.105 | 0.855 | 1.928 | 0.863 | 1.718 | 0.874 | 1.577 | 0.895 |
| | SparseEval | **1.756** | **0.878** | **1.589** | **0.886** | **1.243** | **0.911** | **1.083** | **0.922** | **1.027** | **0.931** |
| Winogrande | Anchor Points | 5.577 | 0.694 | 4.037 | 0.737 | 4.038 | 0.710 | 3.621 | 0.752 | 3.019 | 0.810 |
| | gp-IRT | 4.903 | 0.533 | 3.417 | 0.547 | 2.617 | 0.630 | 2.284 | 0.659 | 1.957 | 0.725 |
| | TailoredBench | 3.212 | 0.705 | 3.740 | 0.751 | 3.478 | 0.760 | 3.130 | 0.752 | 3.120 | 0.788 |
| | SparseEval | **2.088** | **0.794** | **1.500** | **0.853** | **1.361** | **0.867** | **1.182** | **0.882** | **1.027** | **0.897** |

$$\text{AIS}_i = |s_i^\top r|, \qquad i \in A.$$

*Suppose the refinement step removes $i^\star = \arg\min_{i \in A} \text{AIS}_i, j^\star = \arg\max_{j \notin A} \text{CIS}_j$, and forms $A' = (A \setminus \{i^\star\}) \cup \{j^\star\}$. After re-optimizing $w$ over $A'$, the new optimal linear reconstruction error satisfies*

$$E(A') \leq E(A).$$

*Moreover, if $|s_{j^\star}^\top r| > |s_{i^\star}^\top r|$ and $s_{j^\star}$ is not in the linear span of $\{s_i : i \in A\}$, then*

$$E(A') < E(A).$$

## 5 EXPERIMENTS

### 5.1 EXPERIMENTAL SETUP

**Datasets** We collect the LLM evaluation results from Open-LLM Leaderboard (Fourrier et al., 2024), and we obtain model-item accuracy matrix on six LLM benchmarks including ARC (Clark et al., 2018), GSM8K (Cobbe et al., 2021), HellaSwag (Zellers et al., 2019), MMLU (Hendrycks et al., 2020), TruthfulQA (Lin et al., 2022), and Winogrande (Sakaguchi et al., 2021). For LLM benchmarks, it is noteworthy that we expand the number of models from 300 in TinyBenchmark to 5,000, allowing us to more thoroughly evaluate generalization performance across a wider range of models.

**Implementation Details** We randomly select 200 models as the validation and test sets for the LLM benchmarks, with equal sizes. The remaining data are used as the training set. We utilize a 4-layer MLP in LLM Benchmarks and 6e-4 as the learning rate. The refinement step is set to be 10. We utilize MAE and Kendall's $\tau$ as two metrics for evaluation.

### 5.2 MAIN RESULTS

In Table 1, we compare our method with several baselines, including Anchor Points (Vivek et al., 2023), gp-IRT (Polo et al., 2024), and TailoredBench (Yuan et al., 2025). The results demonstrate that SparseEval significantly outperforms all baseline methods, consistently achieving lower estimation error and higher correlation coefficients across varying numbers of anchors. As the number of models scales up to 5000, traditional cluster-based and IRT-based methods struggle to fully utilize the large volume of data. In particular, when using only 100 items for prediction, these methods often exhibit estimation errors exceeding 2%. In contrast, SparseEval leverages anchor refinement

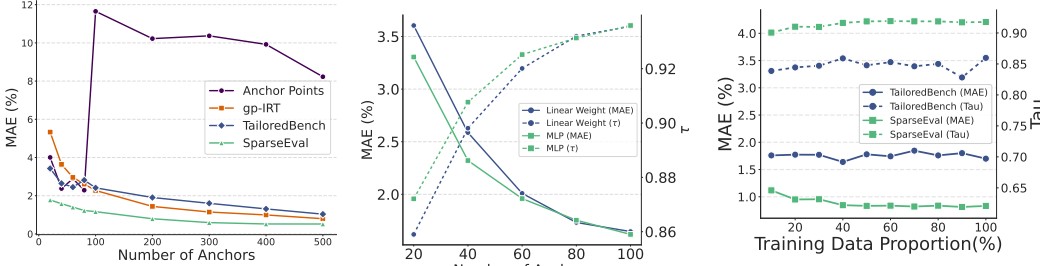

Figure 3: **Error Trend on ARC.** SparseEval consistently outperforms baselines.

Figure 4: **Ablation over network architecture on GSM8K.**

Figure 5: **Ablation over training data proportion on HellaSwag.**

Table 2: **Ablation results on anchor selection.** SparseEval benefits from anchor refinement for downsteam estimates tasks.

| Dataset | Method | Anchor = 20 | | Anchor = 40 | | Anchor = 60 | | Anchor = 80 | | Anchor = 100 | |
|---|---|---|---|---|---|---|---|---|---|---|---|
| | | MAE (%) ↓ | $\tau$ ↑ | MAE (%) ↓ | $\tau$ ↑ | MAE (%) ↓ | $\tau$ ↑ | MAE (%) ↓ | $\tau$ ↑ | MAE (%) ↓ | $\tau$ ↑ |
| ARC | Random | 3.131 | 0.741 | 2.121 | 0.819 | 1.770 | 0.856 | 1.433 | 0.881 | 1.339 | 0.890 |
| | $k$-means | 1.945 | 0.847 | 1.656 | 0.878 | 1.427 | 0.897 | 1.314 | 0.903 | 1.218 | 0.913 |
| | SparseEval | **1.778** | **0.863** | **1.581** | **0.883** | **1.404** | **0.902** | **1.227** | **0.910** | **1.165** | **0.917** |
| GSM8K | Random | 3.831 | 0.856 | 2.694 | 0.896 | 2.467 | 0.908 | 2.086 | 0.918 | 1.857 | 0.928 |
| | $k$-means | 3.544 | 0.860 | 2.491 | 0.898 | 2.042 | **0.925** | 1.767 | **0.933** | 1.631 | **0.938** |
| | SparseEval | **3.305** | **0.872** | **2.321** | **0.908** | **1.960** | 0.925 | **1.754** | 0.931 | **1.619** | 0.936 |
| HellaSwag | Random | 2.465 | 0.728 | 1.804 | 0.788 | 1.389 | 0.839 | 1.207 | 0.858 | 1.152 | 0.867 |
| | $k$-means | 1.631 | **0.859** | 1.258 | 0.882 | **0.973** | **0.906** | **0.931** | 0.909 | **0.772** | **0.919** |
| | SparseEval | **1.477** | 0.857 | **1.210** | **0.890** | 0.993 | **0.906** | 0.942 | **0.910** | 0.827 | 0.918 |
| MMLU | Random | 2.152 | 0.774 | 1.393 | 0.854 | 1.117 | 0.879 | 0.996 | 0.894 | 0.850 | 0.903 |
| | $k$-means | 2.152 | 0.774 | 1.393 | 0.854 | 1.117 | 0.879 | 0.996 | 0.894 | 0.850 | 0.903 |
| | SparseEval | **1.718** | **0.832** | **1.282** | **0.871** | **0.997** | **0.890** | **0.962** | **0.896** | **0.842** | **0.908** |
| TruthfulQA | Random | 2.887 | 0.800 | 2.000 | 0.854 | 1.601 | 0.882 | 1.356 | 0.902 | 1.204 | 0.915 |
| | $k$-means | 2.107 | 0.849 | **1.417** | **0.892** | 1.296 | 0.902 | 1.181 | 0.914 | 1.058 | 0.926 |
| | SparseEval | **1.756** | **0.878** | 1.589 | 0.886 | **1.243** | **0.911** | **1.083** | **0.922** | **1.027** | **0.931** |
| Winogrande | Random | 2.617 | 0.726 | 1.761 | 0.821 | 1.450 | 0.851 | 1.277 | 0.870 | 1.104 | 0.886 |
| | $k$-means | 2.617 | 0.726 | 1.761 | 0.821 | 1.450 | 0.851 | 1.277 | 0.870 | 1.104 | 0.886 |
| | SparseEval | **2.088** | **0.794** | **1.500** | **0.853** | **1.361** | **0.867** | **1.182** | **0.882** | **1.027** | **0.897** |

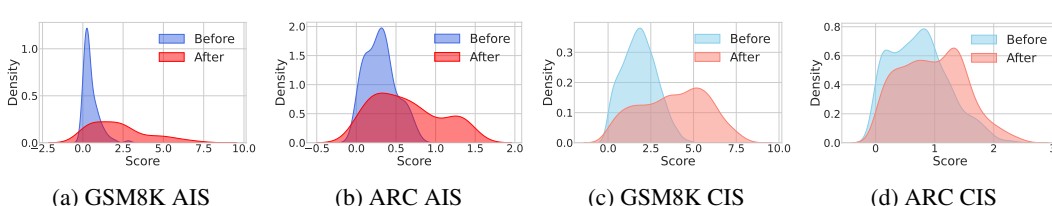

(a) GSM8K AIS      (b) ARC AIS      (c) GSM8K CIS      (d) ARC CIS

Figure 6: **AIS and CIS distribution w.r.t anchor refinement.** The importance scores shift noticeably to the right after refinement, indicating an improvement in the quality of the anchor set.

and weight optimization to fully exploit the training data and enhance generalization via gradient descent, achieving estimation errors below 1%, outperforming prior state-of-the-art approaches by a substantial margin. In addition to a low MAE, SparseEval also achieves Kendall's $\tau$ scores above 0.90, indicating a strong correlation with the original model rankings and further validating the effectiveness and practicality of our method.

## 5.3 ABLATION STUDY

**Over 5x reduction in anchor points than baselines.** As shown in Fig. 3, SparseEval achieves comparable performance to other baselines that use 500 anchors, with only 100 anchors, representing more than a 5× reduction in anchor points. Interestingly, as the number of anchors increases, the performance of Anchor Points drops significantly, indicating that clustering alone without adaptive anchor weight adjustment is far from sufficient.

Table 3: **MAE on the Hold-out Model Families.** SparseEval generalize better on the unseen models.

| Method | TailoredBench | gp-IRT | SparseEval |
|---|---|---|---|
| deepseek-coder-1.3b-instruct | 6.464 | 1.873 | 1.392 |
| deepseek-coder-6.7b-base | 3.751 | 2.083 | 2.132 |
| deepseek-coder-6.7b-instruct | 1.096 | 2.304 | 0.264 |
| deepseek-coder-7b-instruct-v1.5 | 4.182 | 1.681 | 3.700 |
| deepseek-llm-67b-base | 0.020 | 1.882 | 0.730 |
| deepseek-llm-67b-chat | 1.448 | 1.595 | 0.156 |
| deepseek-llm-7b-chat | 2.816 | 2.624 | 0.738 |
| deepseek-math-7b-base | 2.459 | 2.019 | 1.777 |
| deepseek-math-7b-instruct | 3.712 | 1.950 | 3.810 |
| deepseek-math-7b-rl | 3.407 | 2.374 | 2.542 |
| deepseek-moe-16b-base | 0.926 | 2.244 | 1.535 |
| deepseek-moe-16b-chat | 12.865 | 6.195 | 3.740 |
| deepseek-llm-7b-base | 11.662 | 5.710 | 4.616 |
| Max MAE | 12.865 | 6.195 | **4.616** |
| Average MAE | 4.216 | 2.657 | **2.090** |

**Sparse optimization benefits from deeper architecture.** We compare the estimation performance between a traditional linear weight and an MLP-based estimator in Fig. 4. Due to its limited representational capacity, the linear model performs worse, whereas the deeper MLP architecture is better suited for the challenging task of anchor weight optimization.

**Comparable performance with limited training data.** We adjust the proportion of training data and present the results in Fig. 5. As expected, model performance degrades as the amount of training data decreases. However, SparseEval consistently outperforms the baseline, regardless of whether limited or full data is used. Even with only 20% of the data, SparseEval is able to maintain a MAE under 1% and $\tau$ higher than 0.90, demonstrating the robustness of our method.

**Anchor Selection.** We compare different anchor selection methods to investigate the effectiveness of our anchor refinement strategy. As shown in Table 2, anchor refinement clearly leads to a superior anchor set. This indicates that both AIS and CIS, used as metrics for selecting anchors and candidates respectively, are highly aligned with the objective of the downstream prediction task. Notably, the improvement brought by anchor refinement becomes more significant when the number of anchors is limited. This suggests that, in such cases, strategies like $k$-means and random selection fail to correctly represent the characteristics of the original dataset, and multiple rounds of refinement are necessary to discover a better anchor set.

## 5.4 ANALYSIS AND DISCUSSION

**AIS and CIS improvement through refinement.** We further investigate the mechanism behind our anchor refinement process by analyzing the distributions of both anchors and candidates. As can be observed from Fig.6, both AIS and CIS improve significantly after refinement. AIS reflects the sensitivity of the selected anchors to prediction error, while CIS measures how strongly a candidate responds to the predictive behavior of the current proxy model. The consistent improvement of both metrics through refinement indicates a continuous enhancement in the quality of the anchor set.

**Hold-out Families.** To further validate the robustness of our methods on unseen models, we select the DeepSeek family of models as hold-out families. These models span a wide range of scales (from 1.3B to 64B), architectures (both dense and MoE), and application situations (including coder, math, and general models). We train the models on evaluation data from other models and test it on these hold-out models. As shown in Table 3, SparseEval outperforms other baselines in terms of the average and maximum prediction error, demonstrating strong generalization across different models.

**Error Analysis.** We sampled a subset of test examples from both SparseEval and gp-IRT, and visualized their prediction errors, as demonstrated in Fig. 7. We observed that some models evaluated by gp-IRT exhibit errors close to 10%, making it unsuitable for precise prediction tasks. In contrast, SparseEval shows consistently lower average and maximum errors compared to gp-IRT, demonstrating its generality and robustness.

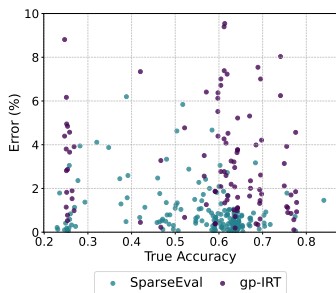

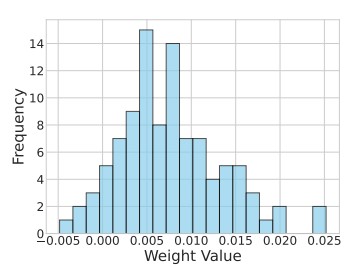

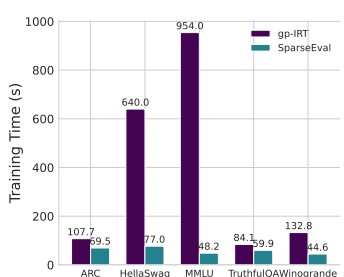

Figure 7: **Error Analysis on MMLU.** SparseEval shows smaller max error than gp-IRT.

Figure 8: **Weight values on MMLU.** Negative value expand the optimization space.

Figure 9: **Efficiency Analysis.** SparseEval is more efficient than IRT-based method.

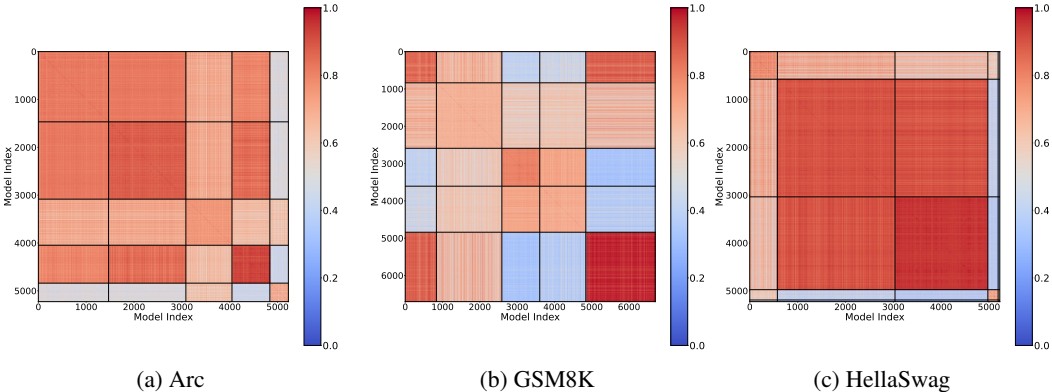

(a) Arc          (b) GSM8K          (c) HellaSwag

Figure 10: **Sparsity in the model-model similarity matrix.** Efficient benchmark can benefit from the similarity and sparsity among models.

**Weight Analysis.** We visualize the distribution of weights obtained from linear weight optimization, as shown in Fig.8. The weight values are relatively concentrated and mostly close to zero. Interestingly, we observe that some weights take negative values that is impossible in traditional cluster-based methods, thereby significantly expanding the optimization space.

**Training efficiency.** We further investigate the efficiency between SparseEval and baselines. The inference cost is based on the number of anchors. After both methods generate anchors and weights, the inference time cost is roughly the same, so we focus mainly on the training time cost. In gp-IRT, the time is mostly spent on training the IRT model. This training process is very time-consuming, taking as long as 16 minutes for training on MMLU. For SparseEval, its time cost is much lower than that of gp-IRT, demonstrating the superior efficiency of our method.

**Model-model similarity.** Surprisingly, we also observe sparsity in the model-model similarity matrix, as shown in Fig. 10. This suggests that the performance of unknown models can potentially be predicted from that of known models, which is consistent with the findings in TailorBench (Yuan et al., 2025). We leave incorporating this sparsity structure into anchor weight optimization and anchor selection in the future work.

## 6 CONCLUSIONS

In this work, we introduce SparseEval, a novel framework that significantly reduces evaluation costs by selecting and weighting a small set of representative test items. We formalize sparse evaluation as a sparse optimization problem and propose the anchor refinement strategy for better anchor selection representativeness. Empirically, SparseEval outperforms baselines with more stable predictions and substantially lower training overhead across various LLM benchmarks. However, our approach has certain limitations since gradient descent requires sufficient samples to guarantee performance. We hope our work could bring inspiration for future work in the field of efficient evaluations.

## ACKNOWLEDGEMENTS

This work is supported in part by the National Natural Science Foundation of China, under Grant (62302309, 62571298).

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

# APPENDIX

## A  DATASET AND LICENSE

We collect the LLM evaluation results of more than 5000 models from Open LLM Leaderboard Fourrier et al. (2024). Below are the datasets used in this paper that have known license information.

The following datasets used in this paper are under the MIT licenses: GSM8K Cobbe et al. (2021) and MMLU Hendrycks et al. (2020).

The following datasets used in this paper are under the CC BY 4.0 licenses: HellaSwag Zellers et al. (2019).

The following datasets used in this paper are under the CC BY-SA 4.0 licenses: ARC Clark et al. (2018).

The following datasets used in this paper are under the Apache-2.0 license: TruthfulQA Lin et al. (2022), Winogrande Sakaguchi et al. (2021).

## B  LLM USAGE

In the preparation of this manuscript, we made limited use of LLMs as a general-purpose writing assistant. Specifically, the LLMs are employed to polish wording, improve sentence fluency, and adjust grammatical structure for clarity and readability. At no point did the LLMs contribute to research ideation, the formulation of hypotheses, methodological design, execution of experiments, or interpretation of results. Their role was strictly confined to surface-level language refinement, comparable to the functions of a grammar-checking or style-editing tool. All intellectual contributions, including the conception of ideas, development of approaches, and analysis of findings, are entirely the work of the authors.

## C  PROPOSITIONS PROOFS

PROOF OF PROPOSITION 1

*Proof.* We begin with the notation defined above. Let $S = [s_1, \ldots, s_n] \in \mathbb{R}^{m \times n}$, and let

$$\mu = SW_a = \frac{1}{n} S \mathbf{1}_n.$$

For an anchor set $A \subseteq \{1, \ldots, n\}$, define

$$\mathcal{W}(A, k) = \{w \in \mathbb{R}^n : \text{supp}(w) \subseteq A, \ \|w\|_0 \leq k\},$$

and the optimal reconstruction error

$$E(A) := \min_{w \in \mathcal{W}(A,k)} \|Sw - \mu\|_1.$$

If $A \subseteq B$, then every feasible weight vector supported on $A$ is also feasible on $B$. Thus

$$\mathcal{W}(A, k) \subseteq \mathcal{W}(B, k).$$

Since minimization is performed over a larger feasible set, we have

$$E(B) = \min_{w \in \mathcal{W}(B,k)} \|Sw - \mu\|_1 \leq \min_{w \in \mathcal{W}(A,k)} \|Sw - \mu\|_1 = E(A).$$

Therefore increasing the number of anchors never increases the optimal error. □

PROOF OF PROPOSITION 2

*Proof.* Since the weights are not at the $L_2$-optimal solution on $A$, the residual $r = Sw - \mu$ is not orthogonal to $\text{span}\{s_i : i \in A\}$, so $s_i^\top r$ contains meaningful variation.

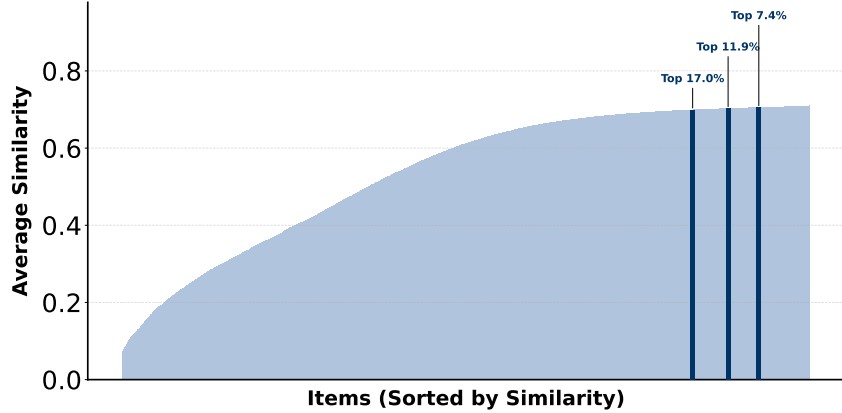

Figure 11: **Representative items analysis on MMLU.** Representative items tends to share high similarity with other items.

For a feature $s_j$, the one-step decrease direction of the linear least-squares loss satisfies

$$\frac{\partial \mathcal{L}}{\partial w_j} = 2 s_j^\top r,$$

so the magnitude $|s_j^\top r|$ measures how much loss can be reduced by adjusting weight $w_j$. Thus CIS identifies the most beneficial candidate to add, and AIS identifies the least beneficial anchor to keep.

Replacing $i^\star$ by $j^\star$ with $|s_{j^\star}^\top r| > |s_{i^\star}^\top r|$ strictly enlarges the set of descent directions for linear least squares. Hence the attainable minimum over the new support $A'$ cannot be larger:

$$E(A') \le E(A).$$

If additionally $s_{j^\star} \notin \mathrm{span}\{s_i : i \in A\}$, then the refinement introduces a new independent descent direction aligned with the residual, yielding a strictly smaller optimal loss:

$$E(A') < E(A).$$

$\square$

## D  ANALYSIS OF REPRESENTATIVE ITEMS

In Figure 8, we observed that certain items have weights with notably high magnitudes, suggesting that they are representative among all items. To better understand these anchors, we compute the average similarity of each item with all other items and highlight those assigned high-magnitude weights. As shown in Fig. 11, these items consistently exhibit high average similarity with other items. This suggests that using them as anchors allows for better representation of other items, which explains why they are assigned larger weights.

Additionally, we also provide the details of these representative items.

---

**Representative Items Samples on MMLU**

The following are multiple choice questions (with answers) about professional law.

...

A man is on trial for rape. The alleged victim testified that she went out to dinner with the man. Afterward, he invited her to his apartment for coffee. Upon entering the apartment, he violently assaulted her. Although she tried to resist, he overpowered and raped her. The man testified that during dinner, he and the victim drank two bottles of Champagne. When they returned to his apartment, he was so intoxicated that he honestly believed that she consented to the intercourse. The jury determined that the victim did not consent to the intercourse. The jury also found that the man, as a result of his intoxication, honestly but unreasonably believed that she was consenting. As a consequence, the defendant should be found
A. not guilty, because he honestly believed that the victim consented.
B. not guilty, because his intoxication negated his criminal intent.
C. guilty, because rape is a general intent crime.
D. guilty, because she did not consent, and his belief that she was consenting was unreasonable.
Answer:

- - - - - - - - - - - - - - - - - - - - - - - - - - - - - - - - - - - - - - - - - - - - - - - - - - - - - - - - - -

The following are multiple choice questions (with answers) about clinical knowledge.

...

Which of the following is true about involuntary movements in the arm?
A. Alcohol makes the tremor of benign essential tremor worse
B. Hemiballismus is due to a stroke causing paralysis of the distal half of the arm
C. A 'milkmaid' grip is sometimes found in dystonia
D. Writer's cramp is an example of a focal dystonia
Answer:

- - - - - - - - - - - - - - - - - - - - - - - - - - - - - - - - - - - - - - - - - - - - - - - - - - - - - - - - - -

The following are multiple choice questions (with answers) about high school physics.

...

During an isothermal expansion, a confined ideal gas does 150 J of work against its surroundings. Which of the following describes the heat transfer during this process?
A. 150 J of heat was added to the gas.
B. 150 J of heat was removed from the gas.
C. 300 J of heat was added to the gas.
D. 300 J of heat was removed from the gas.
Answer:

---

