# OpenReview forum: "SparseEval: Efficient Evaluation of Large Language Models by Sparse Optimization"
_ICLR.cc/2026/Conference — ICLR 2026 Poster_

### Official Review · Reviewer_ky5X · 2025-10-20

**Soundness:** 3
**Presentation:** 3
**Contribution:** 3
**Rating:** 8
**Confidence:** 3

**Summary:**

This paper proposes a framework for faster LLM evaluations that formalizes anchor selection and weighting as an optimization problem. The anchor weights can be learned via an MLP using gradient descent, and the anchors are iteratively updated via candidate importance score and anchor importance score. Across six benchmarks, the SparseEval achieves very low error in terms of MAE and high rank fidelity in terms of Kendall's $\tau$ with very few items (roughly 100) and a large set of models (roughly 5000) from the Open-LLM leaderboard.

**Strengths:**

1. Experiments are comprehensive, results are strong. The study aggregates per-item accuracy matrices over ~5,000 models and evaluates on six widely used datasets, expanding the model coverage far beyond prior efficient-evaluation work (e.g., TinyBenchmarks’ ~300 models). The scaling is nice to see. On this large setting, SparseEval consistently beats baselines (Anchor Points, gp-IRT, TailoredBench) across anchor budgets (20–100). For example, at 100 anchors the method reaches MAE below 1% with $\tau>0.90$ on multiple datasets; the paper highlights that competing methods frequently exceed 2% MAE in the same regime. The error-trend plots (e.g., ARC) show a clear and stable gap over baselines as anchors increase, while the ablation contrasting linear weighting vs MLP underscores that deeper architectures materially improve sparse estimation—precisely the regime SparseEval targets. The data-efficiency ablation is notable: even with 10% training data, the method maintains ≈1% MAE on HellaSwag/MMLU, supporting claims of robustness under scarce supervision.

2. The CIS/AIS refinement loop is intuitive and simple: at its core, the algorithm is greedy by swapping out the anchor with the smallest AIS and in the candidate with the largest CIS. CIS selects candidates whose response patterns align with residual structure; AIS quantifies an anchor’s gradient contribution to error reduction.

**Weaknesses:**

1. Experiments only report MAE and Kendall's $\tau$. Many modern evals use calibration metrics, long-form judgments, or pairwise win-rates. A brief experiment or discussion on extending the loss/aggregator to non-decomposable metrics (e.g., Brier/F1, win-rates) would broaden applicability.

2. Robustness of anchors transfer to newer set of models. The study is comprehensive in-distribution over a very large model set. It would be useful to show that anchors selected on one model cohort transfer to a different cohort or to newer models (e.g., hold-out families), even if with modest degradation.

3. Interpretability of learned weights: The work nicely shows negative weights expand the search space relative to cluster averages; a short qualitative analysis of which items receive high magnitude weights (e.g., skill categories in MMLU) would provide insight for benchmark curation.

**Questions:**

Typos:

* Line 180, $S'=S\odot ({\bf 1}_m^W\top)$ -> $S'=S\odot ({\bf 1}_mW^\top)$.

* Line 185, standard notation for $\ell_1$ norm is $\\|\cdot\\|_1$ rather than $\|\cdot\|_1$.

* Inconsistency on the use of $k$ and k for $k$-means.

---

> ### Author Response · Authors · 2025-11-23
>
> **Q1: Experiments metrics.**
>
> The task we currently focus on is accuracy prediction, and the baselines we compare against follow the same setting. In our method, we use an MLP for prediction, which allows us to directly modify the output layer and train the model with any desired metric as the supervision signal.
>
> However, since all the baselines are designed specifically for accuracy prediction and their methods inherently limit the flexibility of modifying the output, a fair comparison under alternative metrics is not feasible. Therefore, we leave this exploration for future work.
>
> **Q2: Hold-out families experiments.**
>
> Please refer to Q2 in Global Response.
>
> Q3: Interpretability of learned weights.
>
> Please refer to Q3 in Global Response.
>
> **Q4: Typos.**
>
> Thanks for your advice. We have revised the typos in the new version of the manuscript.

---

> > ### Comment · Reviewer_ky5X · 2025-11-24
> >
> > Thank authors for the response, and I don't have further comments. Since my initial score is positive, I'll keep it as is.

---

> > > ### Author Response · Authors · 2025-11-28
> > >
> > > Thank you for your follow-up and for maintaining your positive assessment. We appreciate your time and consideration in reviewing our work.

---

### Official Review · Reviewer_qyxY · 2025-10-23

**Soundness:** 3
**Presentation:** 3
**Contribution:** 3
**Rating:** 6
**Confidence:** 1

**Summary:**

The paper frames large-model benchmarking as a sparse optimization problem: select a small set of anchor items from the full benchmark and learn anchor weights via gradient descent to approximate full-benchmark scores. To improve representativeness, the authors propose iterative anchor refinement using Anchor Importance Score (AIS) and Candidate Importance Score (CIS) to quantify item value; they also use an MLP as a nonlinear aggregator to realize sparse reconstruction. Experiments reportedly achieve low estimation error and high Kendall’s τ ranking consistency across multiple benchmarks, demonstrating substantial cost savings without changing conclusions.

**Strengths:**

The paper offers a clear modeling perspective by framing evaluation sparsity through the model by item score matrix and performing sparse selection with learned weights directly aligned to overall scores and rankings. It introduces task aware anchor optimization via AIS and CIS to iteratively refine anchors beyond one shot selection, and employs a nonlinear MLP aggregator to capture inter item structure and model family differences. Empirically, it foregrounds practical gains, reporting low error and high Kendall's tau across benchmarks, consistent with the aim of reducing cost without altering rankings. Methodologically, it departs from IRT and heuristic clustering and external assessor approaches by explicitly learning anchor weights to reconstruct aggregate benchmark scores.

**Weaknesses:**

1. Do your learned anchors (and the associated aggregation/weights) directly generalize to unseen models? In other words, can we train anchors once and apply them to new model releases without retraining?
2. What is the end-to-end cost of discovering/learning the anchor set (including any proxy-model runs, scoring, and training)? If this cost is substantial, and the resulting anchors/weights only work well for a single model, then the approach may not be worthwhile. Please quantify the cost and clarify the cross-model reusability.
3. I’d like to see visualizations of the item/question space that justify the claimed sparsity (e.g., clustering/affinity structures). Which questions are similar, and why? Please provide empirical evidence (e.g., similarity heatmaps, spectral clustering plots, cluster exemplars) to explain why certain items are considered redundant or representative.

**Questions:**

Same as weakness

---

> ### Author Response · Authors · 2025-11-23
>
> **Q1:  Generalization to unseen models.**
>
> Please refer to Q2 in Global Response.
>
> **Q2: Training cost.**
>
> In fact, we have already provided a comparison with the baseline in terms of training time in Fig. 9 of the manuscript. As demonstrated in Fig. 9, the time cost of SparseEval is much lower than that of the baseline, highlighting the efficiency of our method.
>
> **Q3: Visualization of representative anchors.**
>
> Actually, the sparsity of the model-item matrix is already illustrated in the heatmap of Figure 1, indicating high similarity and cluster structures among items. Additionally, in the weight analysis on MMLU shown in Figure 8, we observed that certain items were assigned significantly larger weights. To further investigate, we visualize these anchors and found that their high representativeness stems from the fact that their prediction vectors can effectively capture and represent those of other items.
>
> Please refer to Q3 in Global Response.

---

> > ### Comment · Reviewer_qyxY · 2025-11-28
> > **Response to authors**
> >
> > I thank the authors for their response. While I am not an expert in this specific area, the problem studied appears to be important, so I am inclined to give a positive assessment. However, my evaluation should be interpreted with the understanding that I have limited expertise in this particular direction.

---

> > > ### Author Response · Authors · 2025-11-28
> > >
> > > Thank you for your positive assessment and for recognizing the importance of the problem we study. We appreciate your careful evaluation, and we fully understand and respect that your expertise may lie in a different area. Your constructive feedback is valuable to us.

---

### Official Review · Reviewer_rZJG · 2025-10-31

**Soundness:** 3
**Presentation:** 3
**Contribution:** 2
**Rating:** 4
**Confidence:** 3

**Summary:**

This paper introduces SparseEval, a novel method to efficiently evaluate large language models by framing the problem as sparse optimization. The method uses a proxy MLP model to learn optimized weights for a small subset of "anchor" test items via gradient descent. A task-aware refinement strategy iteratively improves the anchor set by swapping items based on new "Anchor Importance" and "Candidate Importance" scores. Experiments demonstrate that SparseEval accurately estimates model performance with significantly reduced computational cost, outperforming previous methods in estimation error and ranking correlation.

**Strengths:**

* This paper demonstrates the sparsity in the dataset, which enables the prediction of model performance using a small amount of anchor data.

* The paper proposes SparseEval, which trains an MLP to make predictions based on the performance of existing models on both anchor data and the full dataset.

* Experimental results show that this method can accurately select models that are representative of the entire dataset.

**Weaknesses:**

* Given a new dataset, this method may require testing many models and performing training, which can be costly to train in practice.
* The approach lacks generalization to stronger models and other architectures. Under the current training setup, it remains unclear whether the architecture can generalize to more powerful models or to new architectures such as linear attention or MoE models.
* The method lacks evaluation of its adaptability under long-chain-of-thought (long-CoT) conditions, such as on benchmarks like AIME or GPQA. Under long-CoT settings, model outputs tend to fluctuate significantly, so it is uncertain whether this method would still be effective.
* Training the MLP requires a relatively large amount of data. When only a few existing models are available, the performance may be suboptimal.

**Questions:**

* Could you provide generalization tests of the proposed method, including evaluations on stronger models and models with different architectures?
* Could you provide an analysis of the method’s adaptability on Long-CoT data?
* Could you further explain the sparsity assumption? From Figure 3.1, it appears that the similarity between different clusters of data is also quite high.
* When testing different training checkpoints of the same model, the model’s performance tends to fluctuate — can this method accurately predict the precise performance in such cases?

---

> ### Author Response · Authors · 2025-11-23
>
> **Q1 & Q4: Performance with limited training data.**
>
> Please refer to Q1 in Global Response.
>
> **Q2 & Q5 & Q8: Stronger models and fluctuation on the same model.**
>
> Actually, what we are dealing with is just the model-item matrix, and we only need to obtain the prediction vectors from the models. Therefore, our method can easily generalize to different models. Additionally, regarding the issue of variation across different checkpoints of the same model, we are unable to directly verify this due to the unavailability of intermediate checkpoints during training. However, all such concerns can be generally attributed to evaluating the effectiveness of our method on unseen models.
>
> To address this, we select some models as a held-out family and evaluate the performance of SparseEval and the baselines. Please refer to Q2 in the Global Response for more details.
>
> **Q3 & Q6: Results under long-COT settings.**
>
> Since AIME only contains 30 problems, there is no need for efficient evaluation. Therefore, we conducted relevant experiments on the 448 problems in GPQA. The results are shown in Table A.
>
> The experiments demonstrate that our method can easily generalize to new datasets while achieving superior performance compared to the baselines.
>
> Table A. Performance on GPQA.
>
> |      |               | Anchor = 20 |       | Anchor = 40 |       | Anchor = 60 |       | Anchor = 80 |       | Anchor = 100 |       |
> | ---- | ------------- | ----------- | ----- | ----------- | ----- | ----------- | ----- | ----------- | ----- | ------------ | ----- |
> |      |               | MAE         | tau   | MAE         | tau   | MAE         | tau   | MAE         | tau   | MAE          | tau   |
> | GPQA | gp-irt        | 8.460       | 0.311 | 4.740       | 0.433 | 4.278       | 0.451 | 3.273       | 0.388 | 3.420        | 0.442 |
> |      | TailoredBench | 6.372       | 0.223 | 4.678       | 0.293 | 4.317       | 0.350 | 3.615       | 0.401 | 3.051        | 0.447 |
> |      | SparseEval    | 1.290       | 0.702 | 1.080       | 0.757 | 1.050       | 0.771 | 0.950       | 0.787 | 0.837        | 0.823 |
>
> **Q7: Explanation of the sparsity assumption.**
>
> Since we are working with a model-item prediction matrix composed of -1/1 vectors, and we perform clustering with a fixed number of clusters k=5, it is reasonable for different clusters to exhibit a certain degree of similarity. In fact, it is difficult to determine an exact number of clusters, as the difficulty levels corresponding to different items may not be strictly separable.
>
> However, our preliminary visualization of item-item similarity reveals the sparsity of the original model-item matrix, which manifests in two ways. First, there is intra-cluster similarity present across all datasets, indicating that items within the same cluster can represent each other. Second, in some datasets, we observe high inter-cluster similarity, suggesting that even items from different clusters can contain highly related information. This supports the rationale behind selecting a limited number of anchors to predict the results for all data points.

---

> > ### Comment · Reviewer_rZJG · 2025-11-26
> >
> > Thank you for your response! Since the authors solve most of my questions, I will raise my score.

---

> ### Author Response · Authors · 2025-11-28
>
> Thank you for your positive update and for raising your score toward a positive assessment. We truly appreciate your careful evaluation and are glad that our responses resolved your concerns.

---

### Official Review · Reviewer_cdnC · 2025-11-02

**Soundness:** 3
**Presentation:** 3
**Contribution:** 3
**Rating:** 6
**Confidence:** 3

**Summary:**

The paper proposes SparseEval, a sparse-optimization framework for efficient evaluation of LLMs. It formalizes benchmark sparsity by modeling the model–item score matrix and selecting a small set of representative “anchors.” The method optimizes anchor weights via gradient descent with an MLP predictor and refines anchors using AIS and CIS. Experiments on six LLM benchmarks show that SparseEval achieves <2% MAE and τ > 0.9 with only 100 samples, outperforming prior IRT- and clustering-based approaches.

**Strengths:**

Casting efficient LLM evaluation as a sparse optimization problem is conceptually clean and original.

The gradient-based anchor refinement (AIS/CIS) is simple, intuitive, and empirically strong.

Results across multiple datasets and ablations convincingly show superior efficiency and robustness.

The approach can substantially reduce evaluation cost in large-scale LLM benchmarking.

**Weaknesses:**

The proposed method relies on access to a large model–item performance matrix for training and anchor refinement. Its effectiveness when only a small number of model evaluations are available, or when evaluating a completely new task with limited historical data, has not been examined. This restricts its practical usability in real-world cold-start settings.

While the paper frames efficient evaluation as a sparse optimization problem, the theoretical analysis remains shallow. The notion of sparsity is mainly supported by empirical observations rather than formal proofs or quantitative measures, and there is no theoretical guarantee relating the number of anchors to estimation error.

Typos:
Line 410: demostrating → demonstrating
Line 485: filed → field

**Questions:**

none

---

> ### Author Response · Authors · 2025-11-23
>
> **Q1: Performance with limited training data.**
>
> Please refer to Q1 in Global Response.
>
>
> **Q2: Theoretical analysis.**
>
> Thanks for your suggestion. We have added two propositions in the main text, which theoretically demonstrate that adding anchors can effectively reduce prediction error and support the effectiveness of anchor refinement. Please refer to the manuscript for more details.
>
>
> **Q3: Typos.**
>
> Thanks for your advice. We have revised them in the new version of the manuscript.

---

### Author Response · Authors · 2025-11-23
**Global Response**

# Global Response
**Q1: Performance with limited training data.**

We compare SparseEval with the baseline under different proportions of training data in Fig. 5 in the manuscript. As expected, model performance degrades as the amount of training data decreases, for both the baseline and SparseEval. However, SparseEval consistently outperforms the baseline, regardless of whether full or limited data is used. Even with only 20% of the data, SparseEval is able to maintain a MAE under 1%, demonstrating the robustness of our method.

**Q2: Hold-out family performance.**

To further validate the robustness of our methods on unseen models, we select the DeepSeek family of models as hold-out families. These models span a wide range of scales (from 1.3B to 64B), architectures (both dense and MoE), and purposes (including coder, math, and general models). We train the models on evaluation data from other models and test it on these hold-out models.
As shown in Table 3 in the manuscript, SparseEval outperforms other baselines in terms of the average and maximum prediction error, demonstrating strong generalization across different models.

**Q3: Visualization of representative anchors.**

In Figure 8, we observed that certain items have weights with notably high magnitudes, suggesting that they are representative among all items. To better understand these anchors, we compute the average similarity of each item with all other items and highlight those assigned high-magnitude weights.
As shown in Fig. 11 in the appendix, these items consistently exhibit high average similarity with other items. This suggests that using them as anchors allows for better representation of other items, which explains why they are assigned larger weights.

Additionally, we also provide the details of these representative items.

---

### Comment · Area_Chair_kCZ6 · 2025-11-26
**Reviewer & Author Discussion**

Dear Reviewers,

We kindly encourage you to review and respond to the authors’ rebuttals. Your timely feedback is important for ensuring a fair and thorough review process. Thank you for your contributions to ICLR 2026.

Thank you very much for your time and support.

Best regards,

 Area Chair kCZ6

---

### Author Response · Authors · 2025-12-01
**Rebuttal Summary**

Dear Area Chair,

We are deeply grateful for the time and energy you have dedicated to this process, particularly given the **unique challenges and heavy workload of the current** **ICLR** **cycle**. We sincerely appreciate your time and effort in our work.

To assist with your decision-making, we provide a brief overview of the post-rebuttal status of our reviewers below:

**Reviewer Stance (Post-Rebuttal)**

- `Reviewer cdnC`: While this reviewer was not active in the rebuttal stage, **Reviewer cdnC** has provided **a positive recommendation** from the beginning (**score: 6**).

- `Reviewer rZJG`: Following our detailed responses and additional experiments, **key concerns raised by Reviewer rZJG regarding generalization, adaptability and explanation were effectively alleviated**, leading to an evaluation increase from **4 → 6.** The final score is unfortunately not reflected in the system due to the change of ICLR rebuttal policy, but you may find the expression in the thread below, which state

  > "Thank you for your response! Since the authors solve most of my questions, I will raise my score."

- `Reviewer qyxY`:  **Reviewer qyxY** acknowledged our response and maintained **the positive recommendation** from the beginning (**score: 6**).

- `Reviewer ky5X`:  **Reviewer ky5X** acknowledged our response and maintained **the positive recommendation** from the beginning (**score: 8**).

In conclusion, the reviewers **unanimously lean towards acceptance** following the rebuttal phase. Through the inclusion of comprehensive generalization tests, adaptability analyses (e.g., on new models and datasets), and clarification of the sparsity assumption, **major concerns identified by the reviewers have been comprehensively resolved**. We will ensure that all such enhancements are incorporated into the final manuscript.

Thank you again for your time and for guiding our submission through this unusual review cycle. We hope the above summary helps streamline your decision process.

Warm regards,

The Authors

---

### Meta-Review · Area_Chair_qt98 · 2026-01-05

**Summary:**

Overall, reviewers were generally positive about the paper and viewed SparseEval as a strong and timely contribution to efficient LLM evaluation. They found the core formulation of benchmarking as a sparse optimization problem to be conceptually novel, and highlighted the gradient-based anchor refinement (AIS/CIS) and MLP-based aggregation approach as intuitive, well-motivated, and empirically effective (Reviewers cdnC, qyxY, and ky5X). The main concerns raised by reviewers centered on generalization and applicability in more challenging settings, e.g. cold-start scenarios with limited historical model data, transfer of learned anchors to unseen or newer model families, the cost of learning anchors and training the MLP, long CoT setting experiments, among others (Reviewers cdnC, rZJG, qyxY, and ky5X). These concerns were largely framed as opportunities to further strengthen the proposed approach rather than significant issues, and did not outweigh the reviewers’ overall positive assessment of the paper’s contribution. I tend to agree with their assessment.

**Reviewer Concerns:**

With the additional experiments provided during the rebuttal, most of the major concerns of the reviewers were addressed.

**Reviewer Scores:**

- Reviewer cdnC: The reviewer did not engage with authors during the rebuttal. It is likely they would have either maintained their initial positive score (or increased it slightly) given their concerns were alleviated.

- Reviewer rZJG: The reviewer's concerns were addressed so they would have likely increased the score to reflect a positive assessment.

- Reviewer qyxY: The reviewer decided to maintain their initial positive score.

- Reviewer ky5X: The reviewer decided to maintain their initial positive rating.

---

### Decision · Program_Chairs · 2026-01-26

Accept (Poster)